# Skin lesion classification using multi-resolution empirical mode decomposition and local binary pattern

**Siti Salbiah Samsudin**[1], **Hamzah Arof**[1]*, **Sulaiman Wadi Harun**[1], **Ainuddin Wahid Abdul Wahab**[2], **Mohd Yamani Idna Idris**[2]

**1** Department of Electrical Engineering, Faculty of Engineering, University of Malaya, Kuala Lumpur, Malaysia, **2** Department of Computer System and Technology, Faculty of Computer Science & Information Technology, University of Malaya, Kuala Lumpur, Malaysia

* ahamzah@um.edu.my

**Data Availability Statement:** All images files are availabe from the HAM10000 database. https://dataverse.harvard.edu/dataset.xhtml?persistentId=doi:10.7910/DVN/DBW86T.

## Abstract

Skin cancer is the most common type of cancer in many parts of the world. As skin cancers start as skin lesions, it is important to identify precancerous skin lesions early. In this paper we propose an image based skin lesion identification to classify seven different classes of skin lesions. First, Multi Resolution Empirical Mode Decomposition (MREMD) is used to decompose each skin lesion image into a few Bidimensional intrinsic mode functions (BIMF). MREMD is a simplified bidimensional empirical mode decomposition (BEMD) that employs downsampling and upsampling (interpolation) in the upper and lower envelope formation to speed up the decomposition process. A few BIMFs are extracted from the image using MREMD. The next step is to locate the lesion or the region of interest (ROI) in the image using active contour. Then Local Binary Pattern (LBP) is applied to the ROI of the image and its first BIMF to extract a total of 512 texture features from the lesion area. In the training phase, texture features of seven different classes of skin lesions are used to train an Artificial Neural Network (ANN) classifier. Altogether, 490 images from HAM10000 dataset are used to train the ANN. Then the accuracy of the approach is evaluated using 315 test images that are different from the training images. The test images are taken from the same dataset and each test image contains one type of lesion from the seven types that are classified. From each test image, 512 texture features are extracted from the lesion area and introduced to the classifier to determine its class. The proposed method achieves an overall classification rate of 98.9%.

## Introduction

Pigmented skin lesions, also known as moles or nevus can be malignant or benign [1, 2]. Most skin lesions are benign such as nevus and benign keratosis (BKL), and they are non-cancerous. However malignant lesions are cancerous and they include squamous cell carcinoma (SCC) and basal cell carcinoma (BCC), two of the most common skin cancers. However, among all

**Funding:** the authors received no specific funding for this work.

**Competing interests:** The authors have declared that no competing interests exist.

malignant skin lesions, melanoma is the most dangerous and deadliest [3]. The majority of melanomas arise from pre-existing lesions, hence it is important to observe abnormal changes in lesions that may indicate melanoma transformation [4]. Symptoms that foretell developing melanoma include changes in size, shape and colour. The traditional diagnostic procedure carried out by medical experts is a painstaking task due to the fine-grained variability in the appearance of skin lesions. Thus, an automated system that can detect transformation in lesions and classify their types would surely assist medical experts in expediting the diagnosis.

Recently, deep learning based techniques have received attention from many researchers for skin lesion classification [5, 6]. Anand V. et.al. [7] proposed an Xception model with one pooling layer, two dense layers, one dropout layer and Fully Connected (FC) layer having seven skin lesion classes as the output. They apply data augmentation to HAM10000 dataset to balance the number of images classes with fewer data and obtained 96.4% accuracy. Al Masni et.al [8] compared Inception-v3, ResNet-50, Inception-Resnet-v2 and DenseNet-201 for skin lesion classification of a few datasets and concluded that ResNet-50 performed the best. They also discovered that using a balanced dataset produced a better discrimination performance compared to an imbalanced dataset. Khan M.A et.al [9] proposed a framework that captured skin lesion images using a mobile device and classifying them to several classes. HAM10000 dataset was used with a DenseNet CNN model. The extracted features from fully connected (FC) layers were then downsampled using t-SNE method and fused with those of Multi canonical correlation (MCCA) for classification.

The traditional process of automated skin lesion detection and classification can be divided into 3 main parts namely segmentation, feature extraction and classification. An effective segmentation method locates the area of the lesion and captures its irregular shape by marking its border accurately. Thresholding is an effective segmentation method to use when the lesions have consistent characteristics and the surrounding skin regions are clearly distinguishable from the lesion and homogeneous in nature [2, 10]. However, it produces poor segmentation when the images contain structural, illumination, and color variations, such that no clear threshold that separates the lesion from the surrounding can be found [11, 12]. Active contours is a thresholding method that is robust against noise, artifacts and variations in illuminations and color [13, 14]. It works by minimizing the energy function that contains internal and external terms. The internal terms specify the tension or smoothness of the contour and the external terms guide the contour to move toward the object boundary as the energy of the terms is minimized.

In feature extraction, the characteristics or properties of a lesion are extracted as feature values. These feature values play a vital role in the classification of the lesions as they are used as inputs to a classifier to determine the lesion's type. Different from deep learning that can take the image directly and process it like a black box [6, 15], classical machine learning process is easier to understand and implement. Furthermore, deep learning requires a huge dataset to train. If the dataset contains class imbalance and noise, the classification result may be affected [8, 16]. Normally, GPU is used to train a deep learning classifier to shorten time as it is computationally expensive. Hence, we choose the traditional route of using features and a classifier for our work.

Among all dermoscopic properties of a lesion, color and texture are the most dominant [17, 18]. Grey level co-occurrence matrix (GLCM) is a statistical based texture descriptor that is widely used for skin lesion detection [19–21]. In GLCM, the occurrences of two neighbouring pixels of specific grey levels and directions are counted and stored in a matrix. Several measures can be computed from the co-occurrence matrix and they include entropy, variance, correlation, energy, dissimilarity, mean and run length. Z.Abbas et.al [22] utilized GLCM and Support Vector Machine (SVM) in classifying 3 types of skin lesions and they achieved a

99.02% accuracy on International Skin Imaging Collaboration (ISIC) dataset. R.S.Mahagaon-kar et.al [23] used GLCM and circular shift local binary pattern (CS-LBP) to detect melanoma. They recorded 79.73% and 84.76% classification rates for K-NN and SVM classifier, respectively. M.Elgamal [24] used discrete wavelet transform (DWT) as a feature extractor of skin lesions and selected important features using Principal Component Analysis (PCA). They used the selected features as inputs to an artificial neural network (ANN) or a K-nearest neighbour (KNN) classifier to obtain a 95% and 97.5% accuracy for the respective classifier. Besides DWT and Fourier transform, Empirical Mode Decomposition (EMD) can also be used to decompose a signal [25, 26]. Bidimensional EMD is an extension of EMD that is developed to analyze two dimensional signals like images [27]. Wahba M.A et.al [2] combine features from Bidimensional Empirical Mode Decomposition (BEMD) and grey level difference method (GLDM) to classify malignant BCC and benign Nevus. They used a support vector machine (Q-SVM) as a classier and achieved a perfect result.

BEMD is an adaptive decomposition method suitable for image analysis, but its iterative sifting process is slow and the result is dependent on the content of the image. In an attempt to reduce the sensitivity of BEMD to pixel variations or noise in the image that may cause mode mixing, artificial noise can be distributed randomly in the image before it is decomposed. Although this approach improves the stability of the result, it introduces another step that slows the method even further. Besides, by introducing spurious noise, there will be more extrema to deal with in the image making the surface formation process susceptible to boundary effects, overshoot and undershoot [28, 29]. In this work, we use Multi-resolution Empirical Mode Decomposition (MREMD) with Local Binary Pattern (LPB) to extract texture features of skin lesions. Similar to BEMD, MREMD decomposes an image into several components called Bidimensional Intrinsic Mode Function (BIMF) by sifting process. However, MREMD simplifies and improves the upper and lower envelope estimation to reduce the computation time of the intrinsic mode functions [30]. Our proposed method is different from other BEMDs, as we use a fixed window size for envelope formation in every sifting cycle to reduce the execution time and sensitivity to the number of extrema in the input image. Fixing the window size ensures that results obtained are independent of noise and small variations in the image. Then, LBP is utilized to extract features from the input image and the first BIMF, which produces the most consistent features. Combining features from the input image and the first BIMF improves the classification accuracy of 7 types of skin lesions. Details of the proposed method are given in the following sections.

## Materials and methods

The proposed skin lesion identification consists of three steps, and they are segmentation, feature extraction and classification. The image dataset is mainly obtained from HAM10000 [31] which consist of 10015 total lesion images. The dataset can be grouped into 7 types of lesions namely Actinic Keratosis Intraepithelial Carcinoma (AKIEC), Basal Cell Carcinoma (BCC), Benign Keratosis (BKL), Dermatofibroma (DF), Melanocytic nevus (NV), Melanoma (MEL) and Vascular lesion (VASC). The number images for each class is greatly unbalanced where NV, MEL, BKL, BCC, AKIEC, VASC and DF have 6705, 1113, 1099, 514, 327, 142 and 115 images respectively. To avoid training bias in classes with high number of images [10], we level the number of images for each class to 115. Therefore, for seven classes there are a total of 490 training images and 315 test images. For classes with more than 115 images, the training and test images are selected randomly.

Active contour method is known to be robust against noise and illumination, hence a pre-processing method such as contrast limited adaptive histogram equalization (CLAHE) is not

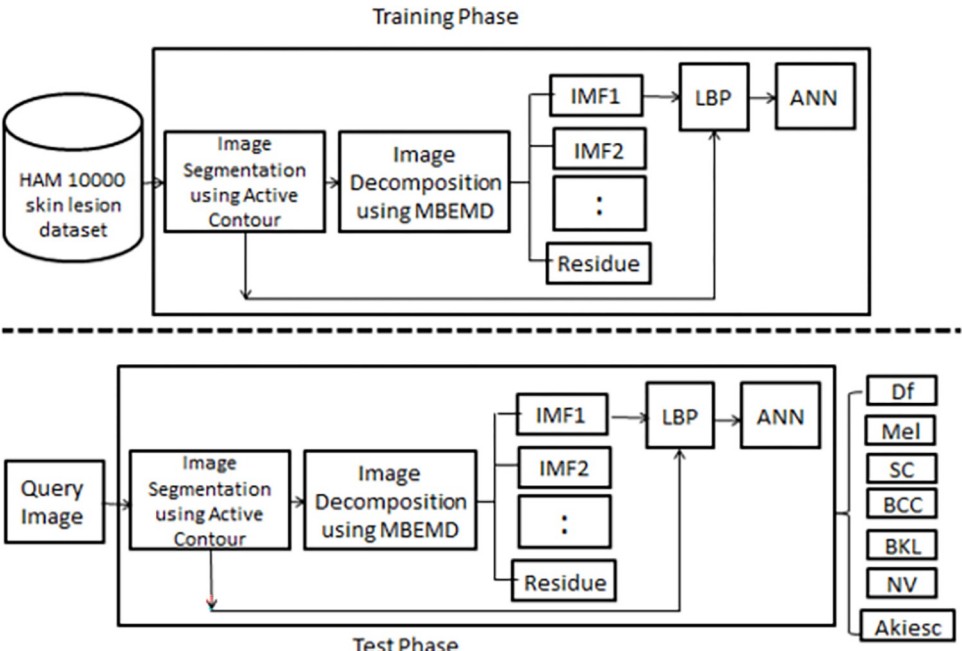

**Fig 1. Overview of the proposed system.**

needed. By skipping the pre-processing step, we reduce the execution time drastically. In segmentation, the lesion area in the input image is segmented using active contour. Next, a copy of the input image is decomposed into several BIMFs using MREMD. However, only the first BIMF is selected to be processed further since it produces the most consistent features with the least intra-class variation of each lesion type. Then LBP is applied to the lesion area in the image and its first BIMF to extract 512 texture features. These features will be used to train an Artificial Neural Network (ANN) classifier and subsequently to classify the test images. The overview of the proposed method is shown in Fig 1 below.

## Active contour

The objective of the segmentation step is to localize the lesion area in the image and separate it from the background. In this paper, we use Chan-Vese active contour model since it is able to detect the lesion boundary without sharp edges and it is also robust to noise [32]. The curve motion in the Chan-Vese model is based on the interaction of the internal and external forces to achieve a minimal energy state when the curve reaches the targeted object boundary. The energy function of the Chan-Vese active contour is given below:

$$F_1(C) + F_2(C) = \int_{inside(C)} |u_0(x,y) - c_1|^2 dxdy + \int_{outside(C)} |u_0(x,y) - c_c|^2 dxdy \qquad (1)$$

where $C$ is a variable curve, $u_0(x, y)$ is the pixel values of input image and constants $c_1$ and $c_2$ are the averages of pixel values inside and outside of $C$ respectively. Chan and Vese [30] noted that the boundary of the object C is the minimizer of the fitting term

$$inf_c\{F_1(C) + F_2(C)\} \approx 0 \approx F_1(C_0) + F_2(C_0) \qquad (2)$$

The above equation explains that if the curve $C$ is outside of the object, then $F_1(C) > 0$ and $F_2(C) \approx 0$. If the curve $C$ is inside the object, then $F_1(C) \approx 0$ but $F_2(C) > 0$. Finally, the fitting

energy equation is minimized if $C = C_0$, when the curve $C$ coincides with the boundary of the object. They also added some regularizing terms as shown below [33]

$$F(C, c_1, c_2) = \mu.length(C) + v.area(insideC) + \lambda_1 \int_{inside(c)} |u_0(x, y) - C_1|^2 dxdy$$

$$+ \lambda_2 \int_{outside(c)} |u_0(x, y) - C_2|^2 dxdy \tag{3}$$

The regularizing terms include the length of curve $C$ and the area inside $C$ where $\mu, v, \lambda_1$ and $\lambda_2$ are fixed parameters and their values are $\mu, v \geq 0$ and $\lambda_1, \lambda_2 > 0$. Finally, when $\lambda_1 = \lambda_2 = 1$ and $v = 0$, the minimization problem can be simplified as

$$inf_{c_1,c_2,c} = F(c_1, c_2, C) \tag{4}$$

## Multi-resolution empirical mode decomposition (MREMD)

MREMD is a variant of bidimensional EMD that is similar to BEMD [34] and FABEMD [35]. However, it is different from them in three aspects. These differences make MREMD faster and less sensitive to the number of extrema in the image. The first difference is MREMD uses a fixed window size for envelope formation of every BIMF to eliminate the time-consuming search for the appropriate window size. Secondly, for the second and higher BIMFs, the size of the input image is reduced by a power of 2 to capture information at a lower resolution. Consequently, the whole process becomes faster but the generated BIMF is also smaller. Upsampling is performed on the BIMF to increase its size back to the original dimension before it is smoothened by an averaging filter. Thirdly, during the envelope formation process, care is taken to ensure that the minima and maxima points coincide with the upper and lower envelope respectively. Details of the sifting process using MREMD are as follows.

The first BIMF is extracted from the input image $I(x,y)$ as follows. Let $Y_1(x,y)$ be a copy of $I(x,y)$. First, $Y_1(x,y)$ is smoothened by average filtering. Then, for each pixel in $Y_1(x,y)$, the maximum and minimum pixel values within a 5x5 window cantered at $(x,y)$, are designated as the upper and lower envelope values at $(x,y)$, respectively. This is repeated for all pixels to form the first upper $U_1(x,y)$ and lower $L_1(x,y)$ envelopes. Next, the upper and lower envelopes are smoothened and averaged to yield the first mean envelope, $M_1(x,y) = (U_1(x,y)+L_1(x,y))/2$. The difference between the input image $I(x,y)$ and the first mean envelope $M_1(x,y)$ is the first BIMF $C_1(x,y)$ defined as $C_1(x,y) = I(x,y)—M_1(x,y)$.

For the second BIMF, we use the first mean envelope $M_1(x,y)$ as the input image $I_2(x,y)$. Let $Y_2(x,y)$ be a copy of $M_1(x,y)$. First, $Y_2(x,y)$ is downsampled by a factor of $2^1$ to reduce its size to $N/2 \; x \; N/2$ and then smoothened by average filtering. Next, $U_2(x,y)$ and $L_2(x,y)$ are formed using the same 5x5 window and smoothened as before. The second mean envelope $M_2(x,y)$ is obtained by averaging $U_2(x,y)$ and $L_2(x,y)$. Since its size is $N/2 \; x \; N/2$, it is upsampled by a factor of 2 using pixel replication to increase its size back to the original dimension. It is smoothened before subtracted from $I_2(x,y)$ to produce the second BIMF, $C_2(x,y) = M_1(x,y)—M_2(x,y)$.

The same steps are repeated to obtain the third and all subsequent BIMFs. In general, for the $n^{th}$ BIMF, the input image is the $(n-1)^{th}$ mean envelope, $M_{n-1}(x,y)$. First, its copy is downsampled by a factor of $2^{n-1}$ and smoothened. Next, the nth upper, lower and mean envelopes $(U_n(x,y), L_n(x,y)$ and $M_n(x,y))$ are obtained using the 5x5 window as before. Then the $n^{th}$ mean envelope $M_n(x,y)$ is expanded by a factor of $2^{n-1}$ by pixel replication to increase its size to $NxN$. Finally it is smoothened before subtracted from the input image $M_{n-1}(x,y)$ to generate the $n^{th}$ BIMF, $C_n(x,y) = M_{n-1}(x,y)–M_n(x,y)$. The number of BIMFs that should be extracted depends on the application, but the extraction process can be repeated until the last mean envelope

Algorithm: Multi-Resolution Empirical Mode Decomposition (MREMD)

Input: grayscale image; $I(x,y)$

Output: BIMFs and residue

Procedure:

1) Set $n=1$ and let $I_n(x,y)$ and $Y_n(x,y)$ be the input image and its copy.

2) Downsample $Y_n(x,y)$ by a factor of $2^{n-1}$

3) Smoothen $Y_n(x,y)$ by a 3x3 averaging filter

4) Find the upper, lower and mean envelope envelopes ($U_n(x,y)$, $L_n(x,y)$, $M_n(x,y)$) of $Y_n(x,y)$

5) Upsample the mean envelope $M_n(x,y)$ by a factor of $2^{n-1}$ by pixel replication

6) Smoothen $M_n(x,y)$ using average filtering

7) Compute the BIMF $C_n(x,y) = I_n(x,y) - M_n(x,y)$

8) Check whether $M_n(x,y)$ is monotonic

**If** false

 Increase $n$ by one and set $I_n(x,y) = Y_n(x,y) = M_n(x,y)$

 Repeat step 2-8

**Else**

 Stop

**Fig 2. MREMD pseudocode.**

becomes monotonic or it becomes too small after downsampling. The summary of the method can be depicted in Fig 2 below and more details of the method are available in [30].

## Local binary pattern

Skin lesion classification is challenging because lesions of different classes may look alike. Besides, within each class there are still variations in size, texture and shape. In this work, only texture features are used from the segmented image since they are the most discriminating. Local Binary Pattern (LBP) is a powerful texture descriptor that is invariant to uneven illumination and rotation. Thus, it is used to extract texture features from the lesion areas of the input image and its first BIMF. For each pixel (at *(x,y)*) in the lesion area, a 3x3 neighbourhood is established. The pixel *(x,y)* is at the center and it is surrounded by eight pixels. The value of each neighboring pixel is compared to that of the center pixel one by one. A label of 1 will be assigned to the pixel if its value exceeds the value of the center pixel, otherwise it will be labelled 0 [36]. After labelling all the eight neighbours, a unique binary code can be formed by appending the eight binary labels around the center pixel in clockwise direction. There are eight possible binary codes that can be formed from appending the eight labels depending on which pixel is chosen as the starting point of the code (its label will be the most significant bit). Normally, the code with the highest value is selected to represent the neighborhood [36].

After an LBP code is assigned to each pixel in the lesion area, a histogram is constructed to record the occurrences of the values of the LBP codes in the area. There are 256 entries to the histogram and their values can be used to represent the textural content of the lesion area.

Since the area of the lesion varies, the entry of the histogram is normalized by dividing them with the area of the lesion. In our work, we use 512 LBP features extracted from the lesion area of the original image and its first BIMF to classify 7 types of lesions.

## Artificial neural network classifier

Artificial neural networks (ANN) classifiers have been widely used in many applications. In this work, a three layer feed-forward ANN which consists of an input layer, hidden layer and an output layer is utilized to classify skin lesion images into 7 classes. The input layer has 512 nodes to cater for the LBP features from the original image and its first BIMF. The hidden layer has 25 neurons while output layer has 7 neurons which matches the number of lesion classes. The number of neurons in the hidden layer is obtained ad-hoc by trial and error. The ANN employs a sigmoid activation function for the hidden layer and a softmax function for the output layer. Initially, the weights are set randomly. Then they are adjusted in the training phase using back propagation algorithm which reduces the error to almost zero.

Other than the number of nodes in the hidden layers, learning rate, momentum and epochs are the hyperparameters that need to be tune while training the data. Learning rate (LR) determine the learning speed, high LR will make the algorithms learn faster but are unstable. While lower LR is more stable but time consuming. Momentum on the other hand, speed up the learning process and help the algorithms not getting stuck in local minima. The tuning of these parameters is done by trial and error, and the initial biases and weights are random numbers between -0.5 to 0.5.

Each feedback loop in the backpropagation process is known as epoch. By setting a higher epoch threshold, the error rate of the training data will be lower as the resulting model fits the training data more perfectly. However, overfitting leads to the model memorizing the training data and not performing well with the test data. To avoid overfitting, we use an early stopping approach to determine the best epoch numbers. First the training epoch threshold is set to 5000 to give plenty of time for the network to fit. Then, by regularly monitoring the error loss of the model performance, the training is stop manually.

## Results

The dataset contained 805 pigmented images of 7 types of skin lesions namely Actinic Keratosis (Akiec), Basal Cell Carcinoma (BCC), Benign Keratosis (BKL), Dermatofibroma (Df), Melanoma (Mel), Melanocytic nevi (Nv) and Vascular lesions (Vasc). These images were randomly chosen from HAM10000 database [31] of the dermatology department of the Medical University of Vienna, Austria and the skin cancer practise of Cliff Rosendahl in Queensland, Australia [31]. For each type of lesion, 70 images were allocated for training and 45 images were reserved for testing. The size of each image was 200x200 pixels. The experiments were implemented using MATLAB R2014a software running on an Intel CORE i7 2.7GHz laptop with 8GB RAM.

Given an image, the process started by segmenting an input image using Chan-Vese active contour to locate the lesion area. Examples of some segmentation results are shown in Fig 3.

Then the image was decomposed into several BIMFs using MREMD. Fig 4 shows the BIMFs and residue of an input image after applying MREMD. It was noted that the first BIMF generated the most consistent LBP features with the least intra class variation. Subsequently, only the first BIMF was extracted by MREMD and used for training the ANN classifier and classification of the test images. As suggested by Bhuiyan et. al. [32] the BIMF was extracted in a single iteration to expedite the sifting step. Using LBP, 256 texture features were obtained from the lesion area of the input image and also from the lesion area of its first BIMF, resulting

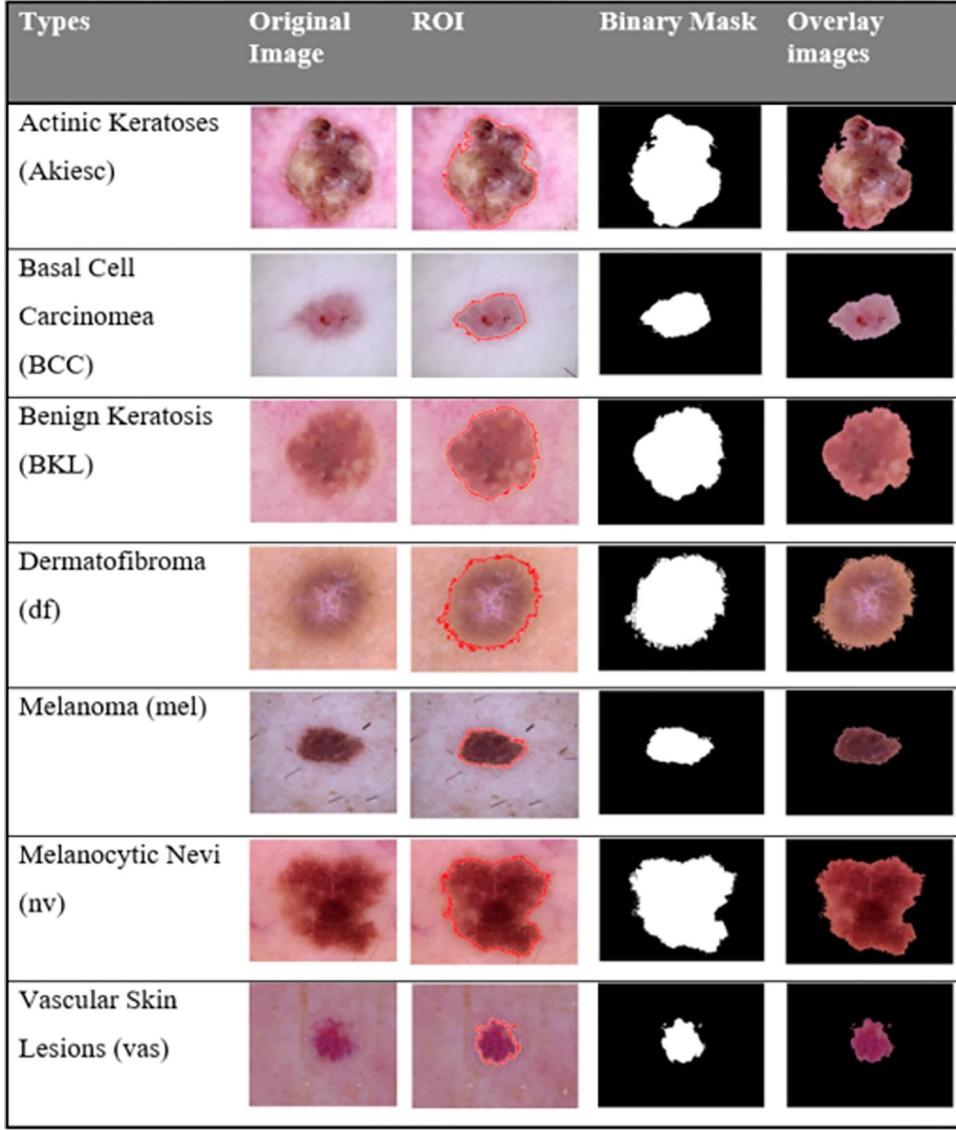

**Fig 3. Samples of segmented skin lesions.**

in a total of 512 features. The process was repeated for each of the training image and test image.

The LBP features from the training images were used to train the ANN classifier. Next, the trained ANN was utilized to classify the LBP features from the test images. The result of the skin lesion classification is shown in the confusion matrix in Table 1 below. The diagonal value represents a true positive and sum values of corresponding column excluding true positive represent false positive. False negative value is the sum of corresponding rows excluding true positive. It is shown that Mel and Vas achieve the highest recognition rate of 100% and BKL and Akiesc has the lowest recognition rate of 93.3%.

The experiment was repeated using the grey level co-occurrence matrix (GLCM) and discrete wavelet transform (DWT) as texture descriptors while keeping the ANN as the classifier. The performance of the LBP features from both the image and its first BIMF was compared to that of the LBP features from the image alone. Performance comparison was also made to the

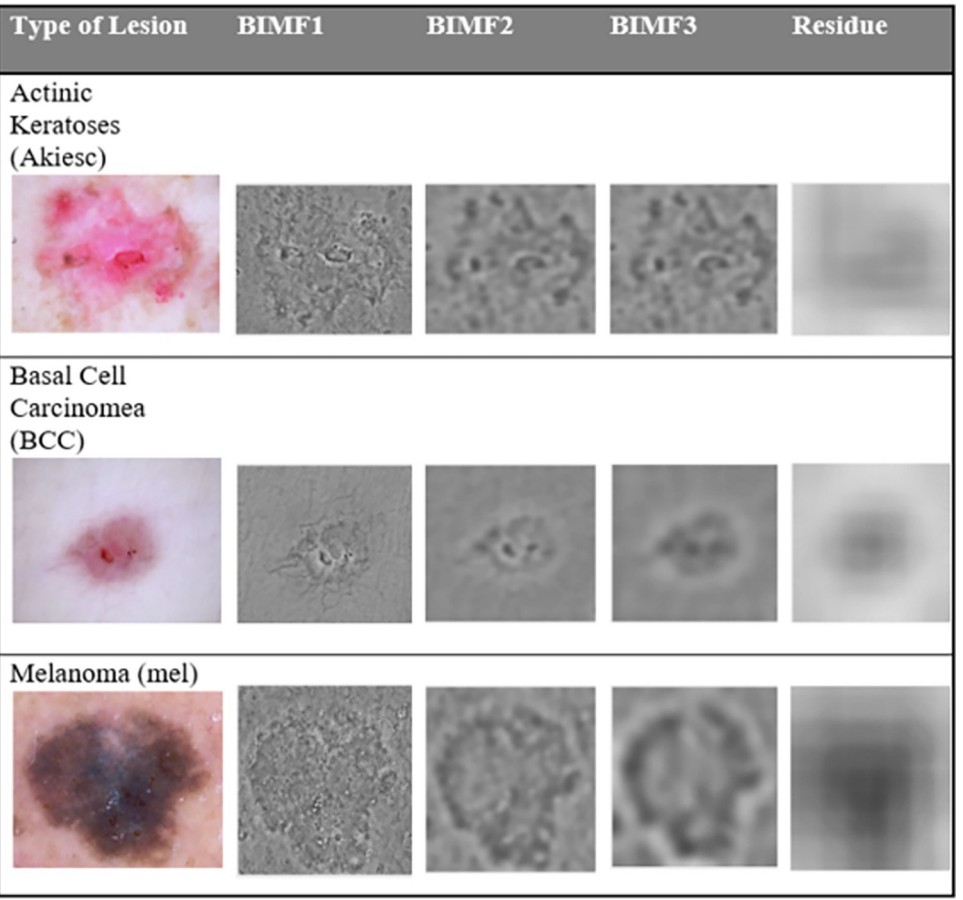

**Fig 4. The decomposed skin lesion using MREMD.**

features of GLCM and DWT in terms of sensitivity, specificity and accuracy as specified below and their relative performances are summarized in Table 2.

$$Sensitivity = \frac{TP}{TP + FN} \tag{5}$$

$$Specificity = \frac{TN}{TN + FP} \tag{6}$$

$$Accuracy = \frac{TP + TN}{TP + TN + FP + FN} \tag{7}$$

**Table 1. Confusion matrix for 7 types of skin lesions.**

|  | Akiesc | BCC | BKL | Df | Mel | Nv | Vas |
|---|---|---|---|---|---|---|---|
| Akiesc | 42 | 0 | 2 | 0 | 0 | 1 | 0 |
| BCC | 0 | 43 | 0 | 0 | 0 | 0 | 2 |
| BKL | 1 | 0 | 42 | 0 | 0 | 2 | 0 |
| Df | 0 | 0 | 1 | 44 | 0 | 0 | 0 |
| Mel | 0 | 0 | 0 | 0 | 45 | 0 | 0 |
| Nv | 0 | 0 | 2 | 0 | 0 | 43 | 0 |
| Vas | 0 | 0 | 0 | 0 | 0 | 0 | 45 |

**Table 2. Comparison of different features extraction methods.**

| Method | Accuracy | Sensitivity | Specificity |
| --- | --- | --- | --- |
| GLCM | 95.3 | 93.2 | 96.1 |
| DWT | 96.8 | 95.4 | 94.8 |
| LBP | 97.4 | 96.2 | 98.4 |
| MREMD + LBP | 98.9 | 96.5 | 99.4 |

Where;

*TP* (True Positive) is the ratio of the number of positive observations to the number of positive true conditions cases,

*FN* (False Negative) is the ratio of the number of negative observations to the number of positive true conditions cases,

*FP* (False Positive) is the ratio of the number of positive observations to the number of negative true conditions cases and

*TN* (True Negative) is the ratio of the number of negative observations to the number of negative observations to the number of negative true conditions cases.

Finally, we replaced the ANN classifier with SVM and KNN to seek improvement to the performance. We found that the ANN still offered the best performance for all texture descriptors. The performance comparison of the three classifiers is depicted in Table 3 and Fig 5 below.

The execution time of three different classifiers for MREMD and LBP as features extractor is shown in Table 4 and Fig 6 below,

## Discussion

HAM10000 dataset contains hugely unbalanced data such that using deep learning for classification tends to favour classes with more images than those with less images. Hence, we equalize the number of lesion images in each class to make it balanced. Therefore, a fair comparison with other techniques that used all or more of the database is not possible. For instance, Huang et. al [37] applied Convolution Neural Network CNN-based method with EfficientNet-B0 as backbone network and achieved 85.8% accuracy. It should be noted that they used the entire database plus their own images. Hemsi et.al [38] used image augmentation by rotating existing images to add data to HAM10000. Using DenseNet201 with CNN method they managed to achieve 87.7% accuracy. Although image augmentation helps balance the number of training data for deep learning, the added data still have a high degree of correlation with the original ones [8]. Another technique to balance the data is using Synthetic Minority Oversampling Technique (SMOTE) [21], but it did not perform well for high dimensionality data and overlapping of classes may occur which generating noisy data. We adopt the traditional route of region segmentation, feature extraction and lesion classification as it is easier to understand. We start by segmenting the lesion area using an active contour method. Then MREMD is applied to the image to obtain its first BIMF. Since the most prominent feature of a skin lesion

**Table 3. Comparison of different classifier methods.**

| Method | SVM | KNN | ANN |
| --- | --- | --- | --- |
| GLCM | 80.6 | 84.9 | 95.3 |
| DWT | 79.4 | 83.2 | 96.8 |
| LBP | 91.4 | 92.6 | 97.4 |
| MREMD + LBP | 95.1 | 96.6 | 98.9 |

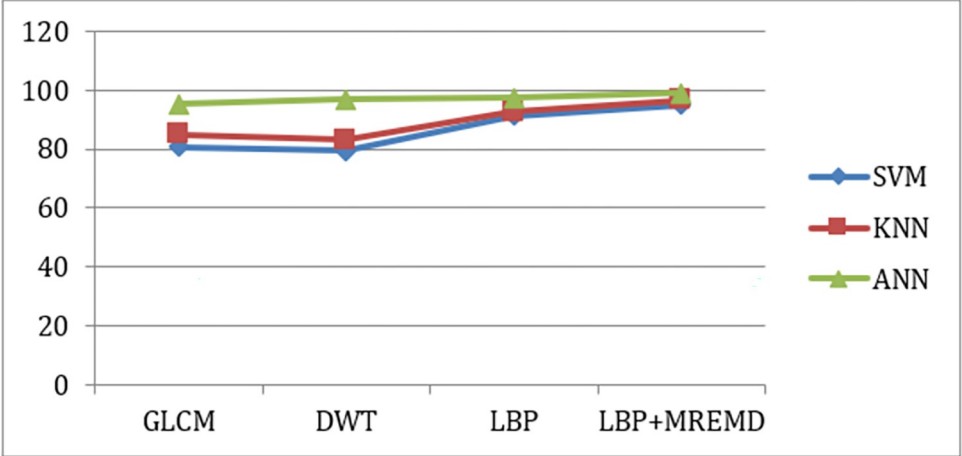

**Fig 5. Performance comparison of different classifiers.**

**Table 4. Execution time for different classifier and feature extractor.**

| Classifier | Feature Extraction Method | Execution time (s) |
|---|---|---|
| SVM | GLCM | 26 |
| | DWT | 23 |
| | LBP | 18 |
| | LBP+MREMD | 25 |
| KNN | GLCM | 35 |
| | DWT | 32 |
| | LBP | 30 |
| | LBP+MREMD | 32 |
| ANN | GLCM | 25 |
| | DWT | 24 |
| | LBP | 20 |
| | LBP+MREMD | 22 |

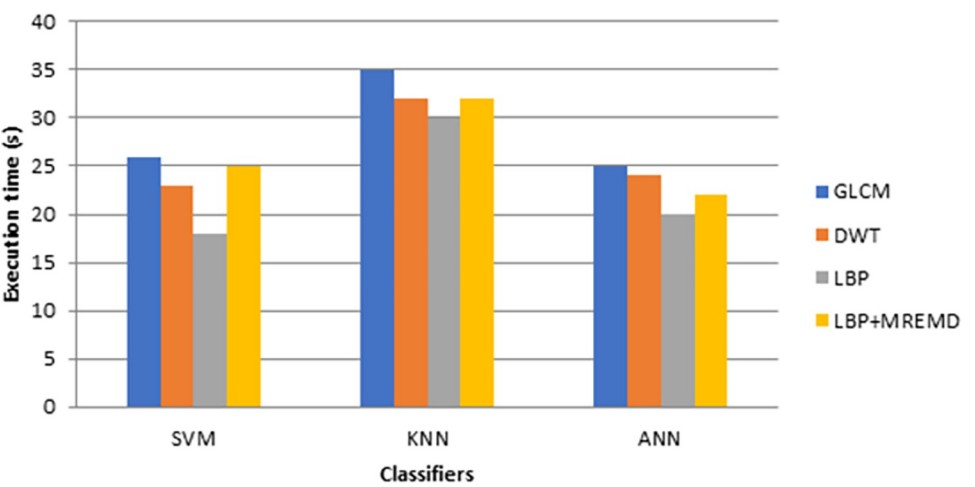

**Fig 6. Comparison of execution time for different classifier.**

is its texture, we use LBP to extract textural features from its image and first BIMF. It is shown that features from our method outperform those from other texture extraction methods such as GLCM and DWT. However, the execution time for our approach is slightly higher that of LBP alone.

## Conclusions

In this paper, a skin lesion classification method that employs LBP, MREMD and ANN is proposed. By combining LBP features from the original image and its first BIMF, the proposed method outperforms the usual approach that employs LBP features alone. Then its performances using two different classifiers are also experimented with. Two other methods of similar complexity namely GLCM and DWT are also tested for comparison. From the experiments, it is shown that adding LPB features from the first BIMF while using ANN as a classifier gives the highest classification rate with 98.9% accuracy.

## Author Contributions

**Supervision:** Sulaiman Wadi Harun, Ainuddin Wahid Abdul Wahab, Mohd Yamani Idna Idris.

**Writing – original draft:** Siti Salbiah Samsudin.

**Writing – review & editing:** Hamzah Arof.

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
