## [Decision Letter · Decision Letter 0]

24 Jun 2022

PONE-D-22-16059Skin Lesion Classification Using Multi-Resolution Empirical Mode Decomposition and Local Binary PatternPLOS ONE

Dear Dr. Arof,

Thank you for submitting your manuscript to PLOS ONE. After careful consideration, we feel that it has merit but does not fully meet PLOS ONE’s publication criteria as it currently stands. Therefore, we invite you to submit a revised version of the manuscript that addresses the points raised during the review process.

We look forward to receiving your revised manuscript.

Kind regards,

Jyotismita Chaki, PhD

Academic Editor

PLOS ONE

Journal Requirements:

2. Please note that PLOS journals require authors to make all data necessary to replicate their study’s findings publicly available without restriction at the time of publication. Please see our Data Availability policy at https://journals.plos.org/plosone/s/data-availability.

As such, please ensure you specify the method used for image selection from the HAM10000 source, such that another researcher would be able to replicate this study. Specifically, please provide detailed instructions regarding which images from each class were discarded to result in 115 images for each class. Alternatively, please make the full specific dataset used in this study (following the selection procedure) available by either A) uploading the full dataset as supplementary information files, or B) including a URL link in your Data Availability Statement and Methods section to where the full dataset (with 115 images of each class) can be accessed.

Additionally, please note that PLOS ONE has specific guidelines on code sharing for submissions in which author-generated code underpins the findings in the manuscript. In these cases, all author-generated code must be made available without restrictions upon publication of the work. Please review our guidelines at https://journals.plos.org/plosone/s/materials-and-software-sharing#loc-sharing-code and ensure that your code is shared in a way that follows best practice and facilitates reproducibility and reuse.

Additional Editor Comments:

Based on the reviewer comments I am suggesting you to revise the manuscript and resubmit.

Reviewers' comments:

Reviewer's Responses to Questions

**Comments to the Author**

1. Is the manuscript technically sound, and do the data support the conclusions?

Reviewer #1: Yes

Reviewer #2: No

2. Has the statistical analysis been performed appropriately and rigorously? 

Reviewer #1: N/A

Reviewer #2: No

3. Have the authors made all data underlying the findings in their manuscript fully available?

Reviewer #1: Yes

Reviewer #2: No

4. Is the manuscript presented in an intelligible fashion and written in standard English?

Reviewer #1: Yes

Reviewer #2: No

5. Review Comments to the Author

Reviewer #1: Comments:

1. The used dataset is very small, it contains only 70 images of each class for training. Would that be enough to justify the results?

2. It is understandable that the authors used a reduced dataset from HAM10000 data for training to balance the classes. However, the testing could have been done on all the available data. Why have they chosen only 315 images for testing?

3. The Novelty/contribution of the paper is not clear. The achieved accuracy is very good but what lead to the improved accuracy, as compared to other papers. Was it the feature extraction, segmentation, or classification method?

4. Figure 1 is not readable. The resolution needs significant improvements.

5. The Resolution of all figures needs improvement.

6. The literature review is weak, Authors should include more recent relevant publications and compare their results with the results of recent publications. For example, these papers should be included

i. Saeed, J. and Zeebaree, S., 2021. Skin lesion classification based on deep convolutional neural network architectures. Journal of Applied Science and Technology Trends, 2(01), pp.41-51.

ii. Javaid, Arslan, et al. "Skin Cancer Classification Using Image Processing and Machine Learning." 2021 International Bhurban Conference on Applied Sciences and Technologies (IBCAST). IEEE, 2021.

iii. Khan, M.A., Muhammad, K., Sharif, M., Akram, T. and de Albuquerque, V.H.C., 2021. Multi-class skin lesion detection and classification via teledermatology. IEEE journal of biomedical and health informatics, 25(12), pp.4267-4275.

iv. Anand, V., Gupta, S., Koundal, D., Nayak, S.R., Nayak, J. and Vimal, S., 2022. Multi-class Skin Disease Classification Using Transfer Learning Model. International Journal on Artificial Intelligence Tools, 31(02), p.2250029.

Reviewer #2: The paper provides nothing new, all proposed methods and data are found in the literature. Moreover, the extracted features are not new and they are too much for the dataset. I think authors should combine these methods with deep learning methods rather than conventional machine learning only.

6. PLOS authors have the option to publish the peer review history of their article (what does this mean?). If published, this will include your full peer review and any attached files.

Reviewer #1: No

Reviewer #2: **Yes: **Ali Mohammad Alqudah

---

## [Author Response · Author response to Decision Letter 0]

13 Jul 2022

Response to reviewer

Reviewer #1: Comments:

1. The used dataset is very small, it contains only 70 images of each class for training. Would that be enough to justify the results?

- The data is sufficient since we are using conventional classifiers rather than CNN with deep learning. Therefore, we believe the results are significant. 

2. It is understandable that the authors used a reduced dataset from HAM10000 data for training to balance the classes. However, the testing could have been done on all the available data. Why have they chosen only 315 images for testing?

- There are many researchers who choose to separate the training and test images. In fact, we hope that separating the training and test images would allow us to see whether the ANN manages to learn the underlying pattern of each class of the training data and then generalize it to the test data that it has not seen before.

3. The Novelty/contribution of the paper is not clear. The achieved accuracy is very good but what lead to the improved accuracy, as compared to other papers. Was it the feature extraction, segmentation, or classification method?

- The novelty comes from using features from the MREMD in addition to features from the traditional LBP to help improve the accuracy. On top of that we also compare the performances of a few different classifiers

4. Figure 1 is not readable. The resolution needs significant improvements.

- Corrected as suggested. 

5. The Resolution of all figures needs improvement.

- Improved as suggested

6. The literature review is weak, Authors should include more recent relevant publications and compare their results with the results of recent publications. For example, these papers should be included

i. Saeed, J. and Zeebaree, S., 2021. Skin lesion classification based on deep convolutional neural network architectures. Journal of Applied Science and Technology Trends, 2(01), pp.41-51.

ii. Javaid, Arslan, et al. "Skin Cancer Classification Using Image Processing and Machine Learning." 2021 International Bhurban Conference on Applied Sciences and Technologies (IBCAST). IEEE, 2021.

iii. Khan, M.A., Muhammad, K., Sharif, M., Akram, T. and de Albuquerque, V.H.C., 2021. Multi-class skin lesion detection and classification via teledermatology. IEEE journal of biomedical and health informatics, 25(12), pp.4267-4275.

iv. Anand, V., Gupta, S., Koundal, D., Nayak, S.R., Nayak, J. and Vimal, S., 2022. Multi-class Skin Disease Classification Using Transfer Learning Model. International Journal on Artificial Intelligence Tools, 31(02), p.2250029.

- Added as recommended

Reviewer #2: 

The paper provides nothing new, all proposed methods and data are found in the literature. Moreover, the extracted features are not new and they are too much for the dataset. I think authors should combine these methods with deep learning methods rather than conventional machine learning only.

- Since the data are limited, it is not suitable to use deep learning for classification since it requires a lot of data to properly train. Furthermore, the classification rates achieved by simple conventional classifiers are high that resorting to deep learning is unnecessary.

---

## [Decision Letter · Decision Letter 1]

7 Sep 2022

Skin Lesion Classification Using Multi-Resolution Empirical Mode Decomposition and Local Binary Pattern

PONE-D-22-16059R1

Dear Dr. Arof,

We’re pleased to inform you that your manuscript has been judged scientifically suitable for publication and will be formally accepted for publication once it meets all outstanding technical requirements.

Kind regards,

Jyotismita Chaki, PhD

Academic Editor

PLOS ONE

Additional Editor Comments (optional):

I am happy to inform you that reviewers are satisfied with the revised version of the manuscript. Therefore I am provisionally accepting the manuscript for publication.

Reviewers' comments:

Reviewer's Responses to Questions

**Comments to the Author**

1. If the authors have adequately addressed your comments raised in a previous round of review and you feel that this manuscript is now acceptable for publication, you may indicate that here to bypass the “Comments to the Author” section, enter your conflict of interest statement in the “Confidential to Editor” section, and submit your "Accept" recommendation.

Reviewer #1: All comments have been addressed

2. Is the manuscript technically sound, and do the data support the conclusions?

Reviewer #1: Yes

3. Has the statistical analysis been performed appropriately and rigorously? 

Reviewer #1: Yes

4. Have the authors made all data underlying the findings in their manuscript fully available?

Reviewer #1: Yes

5. Is the manuscript presented in an intelligible fashion and written in standard English?

Reviewer #1: Yes

6. Review Comments to the Author

Reviewer #1: The authors have addressed all my concerns and I do not have any further objections. The quality of the paper is improved significantly, Therefore, I recommend its acceptance for publication in PLOS ONE.

7. PLOS authors have the option to publish the peer review history of their article (what does this mean?). If published, this will include your full peer review and any attached files.

Reviewer #1: No

---

## [Editor Report · Acceptance letter]

12 Sep 2022

PONE-D-22-16059R1 

Skin Lesion Classification Using Multi-Resolution Empirical Mode Decomposition and Local Binary Pattern. 

Dear Dr. Arof:

I'm pleased to inform you that your manuscript has been deemed suitable for publication in PLOS ONE. Congratulations! Your manuscript is now with our production department. 

Kind regards, 

on behalf of

Dr. Jyotismita Chaki 

Academic Editor

PLOS ONE